# Osteostatin Mitigates Gouty Arthritis through the Inhibition of Caspase-1 Activation and Upregulation of Nrf2 Expression

**DOI:** 10.3390/ijms25052752

**Published:** 2024-02-27

**Authors:** Laura Catalán, María Carmen Carceller, María Carmen Terencio, María José Alcaraz, María Luisa Ferrándiz, María Carmen Montesinos

**Affiliations:** 1Interuniversity Research Institute for Molecular Recognition and Technological Development (IDM), University of Valencia, Av. Vicent Andrés Estellés s/n, 46100 Burjassot, Spain; catalanprades@gmail.com (L.C.); m.carmen.carceller@uv.es (M.C.C.); carmen.terencio@uv.es (M.C.T.); maria.j.alcaraz@uv.es (M.J.A.); 2Department of Pharmacology, Faculty of Pharmacy and Food Sciences, University of Valencia, Av. Vicent Andrés Estellés s/n, 46100 Burjassot, Spain; 3Department of Pharmacy, Pharmaceutical Technology and Parasitology, Faculty of Pharmacy and Food Sciences, University of Valencia, Av. Vicent Andrés Estellés s/n, 46100 Burjassot, Spain

**Keywords:** osteostatin, gouty arthritis, inflammation, caspase-1, immune response, Nrf2

## Abstract

Gouty arthritis results from monosodium urate (MSU) crystal deposition in joints, initiating (pro)-interleukin (IL)-1β maturation, inflammatory mediator release, and neutrophil infiltration, leading to joint swelling and pain. Parathyroid hormone-related protein (107–111) C-terminal peptide (osteostatin) has shown anti-inflammatory properties in osteoblasts and collagen-induced arthritis in mice, but its impact in gouty arthritis models remains unexplored. We investigated the effect of osteostatin on pyroptosis, inflammation, and oxidation in macrophages, as well as its role in the formation of calcium pyrophosphate dihydrate crystals and MSU-induced gouty arthritis in mice models. Osteostatin ameliorated pyroptosis induced by lipopolysaccharide and adenosine 5′-triphosphate (LPS + ATP) in mice peritoneal macrophages by reducing the expression of caspase-1, lactate dehydrogenase release, and IL-1β and IL-18 secretion. Additionally, IL-6 and tumor necrosis factor-α (TNF-α) were also decreased due to the reduced activation of the NF-κB pathway. Furthermore, osteostatin displayed antioxidant properties in LPS + ATP-stimulated macrophages, resulting in reduced production of mitochondrial and extracellular reactive oxygen species and enhanced Nrf2 translocation to the nuclei. In both models of gouty arthritis, osteostatin administration resulted in reduced pro-inflammatory cytokine production, decreased leukocyte migration, and reduced caspase-1 and NF-κB activation. These results highlight the potential of osteostatin as a therapeutic option for gouty arthritis.

## 1. Introduction

Gout is among the most frequent inflammatory arthropathies, characterized by the deposition of monosodium urate (MSU) in the joints and periarticular and subcutaneous structures. Clinical symptoms occur with acute episodes of inflammation, which can lead to persistent clinical manifestations and even become chronic [1,2]. MSU deposition in the synovium leads to reactive oxygen species (ROS) release, resulting in cell death [3]. Additionally, MSU crystals induce the activation of the NLRP3 inflammasome, leading to the caspase-1-dependent cleavage of pro-IL-1β, consequently initiating the release of mature IL-1β from the cell [4]. The presence of MSU crystals can also lead to a significant influx of inflammatory cells, such as monocytes and neutrophils, to the location of MSU crystal deposits [5]. Besides lowering hyperuricaemia, the symptomatic management of acute phases of gouty arthritis comprises anti-inflammatory treatment. However, current therapeutic options, such as colchicine, corticosteroids, non-steroidal anti-inflammatory drugs (NSAID), and IL-1 blockers, are not exempt of limitations and adverse effects, especially colchicine, which has a narrow therapeutic window. Additionally, steroids, such as prednisone, can exacerbate hypertension and diabetes [6].

Parathyroid hormone-related protein (PTHrP) has been shown to be involved in bone generation and restoration. PTHrP can be post-translationally processed to generate several bioactive fragments: an N-terminal fragment (aa 1–36), containing a sequence that activates parathyroid hormone receptor type 1 (PTH1R); three mid-region fragments, involved in calcium mobilization; and a C-terminal fragment (aa 107–139) [7,8,9]. The highly conserved C-terminal region, specifically the penta-peptide sequence Thr-Arg-Ser-Ala-Trp (aa 107–111), known as osteostatin, has been documented to induce bone anabolism and the activation of vascular endothelial growth [10], osteogenic differentiation of mesenchymal stem cells [11], and enhancement of bone regeneration in rat and rabbit models of bone defects [12,13] independently of PTH1R activation. Both the N- and C-terminal domains of PTHrP have been shown to provide protection against the production of reactive oxygen species (ROS) induced by the oxidative stress agent H_2_O_2_ in both murine and human osteoblastic cells [14]. Regarding their inflammatory effects, PTHrP peptides, mainly the C-terminal moiety, have been shown to regulate senescence and inflammation in osteoarthritic osteoblasts, reducing IL-6, PGE_2_, and TNF-α release and COX-2 expression, as well as inhibiting the activation of the NF-κB pathway [15]. Additionally, osteostatin participates in the modulation of osteoclastogenesis through the downregulation of the nuclear factor of activated T cells, cytoplasmic 1 (NFATc1) [16].

We have recently demonstrated the capability of osteostatin to regulate joint inflammation and the degradation of collagen-induced arthritis [17], which displays morphological characteristics akin to rheumatoid arthritis [18]. Given the proved anti-inflammatory and antioxidant properties of osteostatin, together with its beneficial effect on bone homeostasis, this peptide could be a potential candidate to treat acute gouty arthritis. Therefore, we studied the effect of osteostatin decreasing caspase-1 activation and enhancing Nrf2 translocation in activated mouse peritoneal macrophages. Then, calcium pyrophosphate dihydrate (CPPD) and MSU crystal-induced gouty arthritis murine models were used to determine the pharmacologic effects of osteostatin.

## 2. Results

### 2.1. Effects of Osteostatin on Peritoneal Macrophages

#### 2.1.1. Effect of Osteostatin on Cytokine Levels and NF-κB Activation

We explored the regulatory potential of osteostatin on cytokines production using LPS + ATP-stimulated peritoneal macrophages. As shown in Figure 1a, interleukin (IL)-1β, IL-18, TNF-α, and IL-6 levels were significantly decreased by both doses of osteostatin (100 and 500 nM). Subsequently, we investigated the impact of NF-κB regulation on the inhibition of the release of these pro-inflammatory cytokines in peritoneal macrophages by osteostatin. LPS + ATP stimulation induced p65 phosphorylation in these cells, whereas the treatment with 500 nM of osteostatin significantly decreased p65 phosphorylation (Figure 1b,c).

#### 2.1.2. Effect of Osteostatin on Caspase-1 Activation and Lactate Dehydrogenase

Since release of IL-1β and IL-18 requires caspase-1 activation [19], we determined the expression of this enzyme in the supernatants (p20, active form) and cell lysates (p48, procaspase-1) of peritoneal macrophages via Western blotting. Stimulation with LPS + ATP prompted the activation of caspase-1 expressed as the ratio of extracellular p20 and the cell lysate p48. This effect was significantly suppressed by both concentrations of osteostatin (Figure 2a,b). Pyroptosis, a pro-inflammatory type of programmed cell death promoted by the activation of caspase-1, significantly contributes to the onset of inflammatory diseases. After considering the inhibitory effects of osteostatin on caspase-1, we investigated its effect on pyroptosis. We measured lactate dehydrogenase (LDH) release in supernatants of peritoneal macrophages stimulated with LPS + ATP as a marker of pyroptosis. As shown in Figure 2c, LDH increase after LPS + ATP administration was significantly reduced by the highest dose of osteostatin. It is also noteworthy to highlight the absence of the effects of osteostatin on unstimulated macrophages as indicative of its lack of toxicity.

#### 2.1.3. Effect of Osteostatin on Extracellular and Mitochondrial Reactive Oxygen Species (ROS) Production

Inflammatory processes increase oxidative stress, triggering the production of reactive oxygen species (ROS). In Figure 3a,b, stimulation with LPS + ATP caused an increase in fluorescence (red) derived from the oxidation of MitoSOX^TM^ by the generated mitochondrial ROS, which was markedly reduced by 500 nM of osteostatin. To further explore its antioxidant properties, peritoneal macrophages were stimulated with 12-O-tetradecanoylphorbol-13-acetate (TPA), and generated ROS were determined via chemiluminescence. Osteostatin treatment produced a significant ROS reduction, thus confirming its antioxidant effect (Figure 3c). In addition, we tested the highest dose of osteostatin (500 nM) on unstimulated macrophages to rule out a possible pro-oxidant effect of this penta-peptide (Figure 3c).

#### 2.1.4. Effect of Osteostatin on Nrf2 Expression

The control of oxidative stress by Nrf2 may play an important role in the regulation of inflammation and caspase-1 activation. In addition, it has been reported that this Nrf2 transcription factor can inhibit the transcription of inflammatory mediators using mechanisms independent of oxidative stress [20]. Figure 4 shows that in the absence of stimulus, treatment with osteostatin increases the translocation of Nrf2 to the nucleus of targeted cells. In parallel, macrophages stimulated with LPS + ATP also showed a tendency to increase translocation, possibly to control the oxidative response and cell activation induced via this stimulus [21]. Osteostatin treatment further increased Nrf2 translocation.

### 2.2. Effect of Osteostatin on Calcium Pyrophosphate Dihydrate (CPPD) Crystal-Induced Mouse Air Pouch (MAP) Model

The mouse air pouch model simulates an acute, localized inflammatory process, in which mainly macrophages and neutrophils are involved. The formation of the air pouch in the dorsal area of the mouse forms a culture chamber into which the inducing agent, such as calcium pyrophosphate dihydrate (CPPD) crystals, can be injected, which activates leukocyte migration, leading to the release of enzymes and inflammatory mediators [22].

#### 2.2.1. Effect of Osteostatin on Cell Migration, Myeloperoxidase (MPO) and Cytokines Release on Air Pouch Exudates

CPPD crystals induced cell activation and influx to the air pouch at 6 h after stimulation; in this experimental model, migrating cells are predominantly neutrophils [22]. Figure 5a shows that the intra-pouch administration of both osteostatin concentrations significantly reduced the number of infiltrating cells in air pouch exudates. Upon activation by CPPD, neutrophils released granule contents, enhancing myeloperoxidase (MPO) activity in pouch exudates, which is markedly reduced by both osteostatin concentrations (Figure 5b). These results demonstrate that osteostatin exerts inhibitory effects on cell migration and neutrophil degranulation. Figure 5d shows how the cytokines IL-1β and IL-18, both related to the activation of caspase-1, experience a significant decrease in their concentration in the air pouch in the groups of mice treated with osteostatin, obtaining 58% inhibition for IL-1β and practically 100% inhibition for IL-18 at a dose of 6 μg/pouch. For the same dose, a marked decrease in the concentration of TNF-α and IL-6 is also observed, reaching inhibition values of 90% and 100%, respectively. Likewise, CXCL-1 chemokine levels were strongly reduced in the groups treated with osteostatin (Figure 5c).

#### 2.2.2. Effect of Osteostatin on Caspase-1 and NF-κB Activation

Given the reduction in IL-1β and IL-18 cytokine levels in the air pouch exudate of the osteostatin-treated mice, we determined caspase-1 activation in this model by immunoblotting the active p20 subunit of caspase-1 in exudate supernatants and the procaspase-1 (p48) in the cell pellet lysates. Figure 6a,b showed a strong reduction in p20/p48 at the two doses of osteostatin tested, confirming the inhibitory effect of this peptide on caspase-1 activation in this model.

CPPD crystal injection into the air pouch induced the phosphorylation of the p65 subunit of NF-κB in infiltrating cells, which was inhibited via osteostatin administration. This subunit is essential for transporting active complexes into the nucleus, implying an inhibition of p65 acetylation and subsequent NF-κB activation. The downregulation of NF-κB activation stands out as a potential mechanism underlying the anti-inflammatory effect of osteostatin.

### 2.3. Effect of Osteostatin on Monosodium Urate (MSU) Crystal-Induced Gouty Arthritis Model

The next in vivo model chosen to study the anti-inflammatory impact of osteostatin is the gouty arthritis model induced via the subcutaneous injection of monosodium urate (MSU) crystals [23]. The precipitation of these crystals is a key feature of gout [19].

#### 2.3.1. Effects of Osteostatin on Plantar Edema and Inflammatory Mediators

The severity of gouty arthritis exhibited a continuous increase throughout the experiment (24 h). However, mice treated with osteostatin (80 µg/kg or 120 µg/kg, subcutaneous) demonstrated a consistent decrease in this severity score, with the most significant improvement observed at 24 h for both concentrations (Figure 7a,b). MSU crystal injection induced increases in IL-1β, IL-18, TNF-α, IL-6, and CXCL-1, and the subcutaneous administration of osteostatin significantly inhibited all these cytokines (Figure 7c). Notably, there were decreases in IL-1β and IL-18 (83% and 100% inhibition, respectively, at the dose of 80 μg/kg), both associated with caspase-1 activation. In gouty arthritis, neutrophils are recruited and attempt to phagocyte urate crystals, which can be oxidized via MPO [24]. MPO activity was determined in paw homogenates after 24 h of MSU crystal administration. MSU crystals enhanced MPO activity, whereas both concentrations of osteostatin decreased this mediator activity (Figure 7c). The significant reduction in the chemotactic factor CXCL-1 levels aligns with the inhibition of MPO activity observed in the previous results, supporting the inhibitory effect of osteostatin on neutrophil infiltration (Figure 7c).

#### 2.3.2. Effects of Osteostatin on Caspase-1 and NF-κB Activation

As shown in Figure 8a,b, MSU crystals lead to an increase in the p20 subunit signal in control mice compared to unstimulated mice. In the osteostatin-treated animals, a significant decrease in this signal is observed, suggesting a decrease in caspase-1 activity and, therefore, a mitigated inflammatory response. These findings agree with those presented in Figure 7c, demonstrating that osteostatin reduces the production of the active cytokines Il-1β and IL-18. Therefore, the anti-inflammatory effect of osteostatin in the gouty arthritis model would be, in part, associated with the inhibition of caspase-1 activation induced by MSU crystals. In the inflammatory response produced by MSU crystals in the joint cavity, the involvement of transcription factor NF-κB has been demonstrated [25]. As depicted in Figure 8c,d, the p65 phosphorylated fraction of NF-κB increased in the control group compared to the naïve group, indicating the activation of this transcription factor. Notably, in the osteostatin-treated groups, there was a reduction in phosphorylation at both tested doses.

## 3. Discussion

The outcomes of this study have provided further insight into the pharmacological properties of the PTHrP (107–111) C-terminal peptide osteostatin, which alleviates the acute oxidative and inflammatory processes in both crystal-induced pseudogout and gouty arthritis models. Our research group had already shown that osteostatin modulates osteoclastogenesis [15] through NFATc1 inhibition [16], as well as its anti-inflammatory potential in osteoarthritic osteoblasts [15] and collagen-induced arthritis in mice [17]. In this model, osteostatin reduced the local production of the pro-inflammatory cytokines IL-1β, IL-6, IL-17, and TNF-α, which are increased in patients with arthritis and enhanced the release of the anti-inflammatory cytokine IL-10. In addition, it reduced cellular infiltration in the joint cavity [17]. The anti-inflammatory profile of the osteostatin has been confirmed in the present study, using the ex vivo mouse peritoneal macrophage model, and the CPPD crystal-induced pseudogout and MSU-induced gouty arthritis models.

In agreement with the scientific literature, mouse peritoneal macrophages were stimulated with LPS (as a priming signal to activate TLR4 receptors) and ATP (activator of the NLRP3/caspase-1 system through P2X7 purinergic receptors binding) [26]. In LPS + ATP-stimulated macrophages, osteostatin reduced the release of the cytokines IL-1β, IL-18, TNF-α, and IL-6, which play an important role in numerous inflammatory pathologies. This effect could be related to the reduction in NF-κB and caspase-1 activation via osteostatin treatment, as previously reported in osteoarthritic osteoblasts [15]. NF-κB is part of the priming signal for the transcriptional activation of the inflammasome and pro-IL-1β, which would subsequently lead to the second phase of caspase-1 activation [19]. Also, IL-1β released by caspase-1 can contribute, in turn, to the activation of NF-κB by activating its specific receptors [27]. In addition, IL-1β and IL-18 are cytokines that depend on the activation of the inflammasome and caspase-1 for their active secretion. Thus, our results suggest that caspase-1 inhibition may contribute to the inhibitory effects of osteostatin on IL-1β and IL-18 levels.

In pseudogout and gouty arthritis, there is an excess production of reactive oxygen species (ROS) and pro-inflammatory cytokines. Although the role of ROS and mitochondria in NLRP3 inflammasome activation remains controversial, the use of antioxidants blocks NLRP3-dependent caspase-1 activation, indicating that redox signaling or oxidative stress is involved in this process [19]. Several authors suggest that ROS participate in the initial signal necessary for NF-κB-mediated NLRP3 transcription [25,28]. Others give greater interest to the activation of NOX, or mitochondrial disturbance, since most of the stimuli that participate in the activation phase of the inflammasome increase mtROS [29,30]. Our in vitro assays indicate that osteostatin reduced ROS production in macrophages, mainly at the mitochondrial level. These results may confirm the inhibitory effects of this molecule on oxidative stress, as previously demonstrated in osteoblastic cells [14]. The decrease in ROS generation could lead to a reduction in NF-κB activity and osteostatin-produced caspase-1 activation. Additionally, osteostatin may be involved in the protective effects on cell viability, as demonstrated in this study.

Regarding its antioxidant effect, treatment with osteostatin increased the translocation of Nrf2 to the cell nucleus. In basal conditions, the transcription factor remains inactive in the cell cytoplasm bound to the protein inhibitor Keap1, but when the cell is exposed to oxidative stress, it dissociates and translocates inside the nucleus to activate the transcription of antioxidant, anti-inflammatory, and cell survival genes [21]. In addition, Nrf2 can inhibit the transcription of inflammatory cytokines independently of oxidative stress [20]. Several studies indicate that Nrf2 activation has a negative regulatory effect on the activation process of the NLRP3 inflammasome and the release of the mature form of IL-1β by caspase-1 through the expression of the cytoprotective enzyme NQO1 [31]. Other authors demonstrate that the translocation of Nrf2 to the nucleus inhibits the expression of NLRP3 at the transcriptional level by inhibiting the activation of NF-κB, suggesting a clear inter-relationship between these signaling pathways [32].

In gouty inflammation, the release of IL-1β produced via the activation of the inflammasome plays a key role; thus, we decided to approach an animal model of gouty arthritis induced by MSU crystals to mimic the characteristics of human gouty arthritis [33]. The activation of macrophages by MSU crystals induces NRLP3 activation, leading to caspase-1 activation and the subsequent release of IL-1β and IL-18, among other pro-inflammatory cytokines [23,34]. These cytokines and the crystals themselves activate the different cell types present in the area (macrophages, endothelial cells, fibroblasts, etc.), resulting in the release of numerous inflammatory mediators such as chemokines and cytokines. The production of IL-8 (in humans) or CXCL1 (in mice) promotes the infiltration of neutrophils to the crystal focus and their activation, leading to further inflammatory mediator release that amplifies the response, resulting in edema and pain [35]. In addition, in inflammation induced by CPPD crystals, the important role of IL-1β in the migration of neutrophils in the affected area has been demonstrated [36].

In both in vivo models, CPPD and MSU crystals induced the production of IL-1β in the inflammatory focus, in which the production of the chemokine CXCL-1 led to intense neutrophilic migration. In both models, osteostatin was shown to have anti-inflammatory properties, controlling both the vascular and cellular phases of the response. Osteostatin effectively decreased the joint swelling in the MSU crystals model. The inhibition of cell migration would depend on the reduced production of CXCL-1, which, in turn, is related to lower levels of IL-1β in the inflammatory focus. In agreement with the in vitro assays, decreased activation of caspase-1 could be responsible for the control of IL-1β and IL-18 levels. However, the activation of pro-IL-1β in the inflammatory response may occur via other mechanisms independent of caspase-1, for example, neutrophil elastase activity [37], which could be responsible for a more pronounced effect of osteostatin on pro-IL-18 activation.

The inflammatory response to microcrystals is known to be NF-κB-dependent. Thus, the activation of NF-κB via CPPD or MSU determines the production of IL-1β, IL-18, IL-6, TNF-α, and CXCL-1 [22]. As observed in both in vivo models, osteostatin reduced the activation of NF-κB induced by crystals, and consequently, the decreased production of these mediators resulted in the attenuation of the inflammatory process. The results of the in vitro, using mouse peritoneal macrophages, and in vivo studies suggest that osteostatin could regulate the first phase of priming by inhibiting the activation of NF-κB and, consequently, the transcription of pro-inflammatory cytokines. Additionally, osteostatin may influence the subsequent phase of inflammasome complex formation, thereby reducing the caspase-1 activation and, subsequently, the release of the active forms of IL-1β and IL-18. However, further studies are required to precisely elucidate the underlying mechanism of action responsible for the effects of osteostatin.

Our results confirm that osteostatin has emerged as a promising anti-inflammatory candidate in the treatment of acute gouty arthritis. This penta-peptide presents the advantage of lower MW and, therefore, less immunogenicity, than anti-IL-1 biological therapies. Moreover, another problem with current anti-inflammatory treatments used during the acute gouty attacks, such as colchicine, is its high toxicity [6]. In this regard, it is noteworthy that after s.c. daily administrations of osteostatin over 15 days, there was no visible sign of toxicity or behavioral change in the chronic murine arthritis model induced by Collagen II [17]. Nevertheless, we have to acknowledge that further studies are necessary to demonstrate the translatability of our results in animal studies to the clinic. Additionally, future investigations will be performed to further explore the mechanism involved in caspase-1, NF-κB, and Nrf2 pathways using human macrophages isolated from whole peripheral blood.

In conclusion, our results show that osteostatin downregulates the acute inflammatory response in gouty arthritis based on its favorable effect inhibiting caspase-1 and NF-κB activation, as well as promoting Nrf2 translocation, proving its potential interest as a new strategy for the development of future therapies in joint diseases.

## 4. Materials and Methods

### 4.1. Animals

Male C57BL/6 mice (Charles River, Écully, France) between 10 and 12 weeks of age (20–25 g) were used for all the experiments. Mice were maintained at 21 ± 2 °C on a 12 h light–dark cycle with feed and water ad libitum in the housing facility of the School of Pharmacy of the University of Valencia. All experiments were performed following the European regulations for the handling and use of laboratory animals with the corresponding approvals and authorizations. Corrective measures were implemented systematically to minimize any potential suffering experienced by the animals under study. Various parameters were closely monitored, with values ranging from 0 to 3, covering aspects such as the animal’s posture, coat condition, eye/nasal secretions, aggressiveness during handling, and vital signs, including changes in body temperature and heart rate. This evaluation also included an assessment of spontaneous behavior, encompassing inactivity, self-mutilation, abnormal vocalizations, and weight loss, the latter being designated as 3 when exceeding 20% of the body weight loss. Based on the cumulative score assigned to each animal, suggested corrective actions were outlined as follows: a score of 0–4 indicated a state of normalcy, 5–9 warranted careful supervision, and a cumulative score of 10–20 signified severe suffering, prompting consideration of euthanasia as an ethical intervention. At the end of the different procedures, animals were anesthetized with 4–5% isoflurane in a SomnoSuite (Kent Scientific, Torrington, CT, USA) and euthanized via cervical dislocation.

### 4.2. Isolation and Culture of Peritoneal Macrophages

To isolate elicited macrophages from the peritoneal cavity of C57BL/6 mice, 1 mL of 3% Brewer thioglycolate medium (#Cat. T-9032; Sigma-Aldrich, St. Louis, MO, USA) in water was injected intraperitoneally. After 96 h, elicited cells were harvested with 5 mL of phosphate-buffered saline (PBS) (#Cat. 10010-015; Gibco, Life Technologies limited, Paisley, UK) and centrifuged at 400× *g* for 6 min. Then, the pellet was resuspended in RPMI 1640 (Roswell Park Memorial Institute Medium,#Cat. L0498; Biowest, Riverside, MO, USA) medium at 37 °C, and cells were seeded at 2 × 10^6^ cells/mL in RPMI supplemented with 10% Fetal Bovine Serum (#Cat. S181B-500; Biowest^®^, Riverside, MO, USA) and 1% penicillin/streptomycin (#Cat. 15140-122; Gibco, Life Technologies Corp., Grand Island, NY, USA). Cells were maintained under standard culture conditions (5% CO_2_-enriched atmosphere at 37 °C) for 18h [38]. Before each experiment, the medium was replaced, and then cells were treated with osteostatin (OT). Osteostatin (1–5) amide trifluoroacetate salt (#Cat. 4025761.0025; Bachem, Bubendorf, Switzerland) was first dissolved in saline solution to achieve a concentration of 10 µM, from which subsequent dilutions of 100 nM and 500 nM were prepared in the culture medium to conduct macrophage studies. Macrophages were incubated with 100 nM and 500 nM of osteostatin for 30 min and stimulated with 1 µg/mL of lipopolysaccharide (LPS) (#Cat. L4391; Sigma-Aldrich, St. Louis, MO, USA) for 4 h (priming). Afterwards, the medium was changed to RPMI without FBS, and primed cells were stimulated with 5 mM adenosine 5′-triphosphate (ATP) (#Cat. A3377; Sigma-Aldrich, St. Louis, MO, USA) for 10 or 30 min.

### 4.3. LDH Assay

Pyroptosis induced via caspase-1 activation was assayed through the determination of the lactate dehydrogenase (LDH) activity in supernatants of elicited macrophages treated with osteostatin in the absence or presence of LPS (1 μg/mL) + ATP (5 mM) stimulus. Then, 50 µL of supernatant and 50 µL of solution A, composed of 0.2M Tris buffer, pH 7.2, 250 µg β-nicotinamide adenine dinucleotide (β-NAD) (#Cat. N6005; Sigma-Aldrich, St. Louis, MO, USA), 1.2 mg lactic acid (#Cat. L1375; Sigma-Aldrich, St. Louis, MO, USA), 130 µg thiazolyl blue tetrazolium (MTT) (#Cat. M2128; Sigma-Aldrich, St. Louis, MO, USA), and 30 µg of phenazine methosulphate (#Cat. P9625; Sigma-Aldrich, St. Louis, MO, USA), were added to a 96-well plate [39]. Then, the cells were incubated for 90 min at 37 °C in the dark. Absorbance was measured at 550 nm using a Wallac 1420 VICTOR3^TM^ microplate spectrophotometer (PerkinElmer, Turku, Finland).

### 4.4. Enzyme-Linked Immunosorbent Assay (ELISA)

The elicited macrophages at 10^6^ cells/mL in 6-well plates were incubated with osteostatin in the absence or presence of LPS (1 μg/mL) + ATP (5 mM) stimulus. TNFα, IL-1β, IL-18, IL-6, and CXCL-1 were measured in supernatants with sandwich ELISA kits with a sensitivity of 31.3 pg/mL for TNFα (#Cat. DY410-05; R&D Systems, Minneapolis, MN, USA), 15.6 pg/mL for IL-1β (#Cat. DY401-05; R&D Systems, Minneapolis, MN, USA), 19.0 pg/mL for IL-18 (#Cat. BMS618-3; Thermo Fisher Scientific, Göteborg, Sweden), 4.0 pg/mL for IL-6 (#Cat. 88-7064-88; Thermo Fisher Scientific, Waltham, MA, USA), and 4.0 pg/mL for CXCL-1 (#Cat. DY453-05; R&D Systems, Minneapolis, MN, USA) according to the manufacturer’s instructions using a Wallac 1420 VICTOR3^TM^ microplate reader (PerkinElmer, Turku, Finland).

### 4.5. Western Blotting

The protein concentrations from supernatants and cell lysates of peritoneal macrophages, air pouch exudates, and paw homogenates were determined using the DC Bio-Rad Protein assay kit (Bio-Rad, Hercules, CA, USA). Cell supernatants were used to determine the active p20 fraction of caspase-1 in macrophages and air pouch exudates. Adherent macrophages were harvested with 500 µL of lysis buffer. After centrifugation at 10,000× *g* for 10 min at 4 °C, supernatant cell lysate was used to determine the protein expression of procaspase-1 (p48) and the phosphorylated p65 subunit of NF-κB. Proteins (10–20 µg/lane) were separated with the use of sodium dodecyl sulfate-polyacrylamide gel (SDS-PAGE) electrophoresis (12.5% for caspase-1 and 10% for NF-κB) and transferred into polyvinylidene difluoride (PVDF) membranes (GE Healthcare Life Sciences, Barcelona, Spain). Membranes were blocked with 3% (*w*/*v*) non-fat dry milk and incubated with specific antibodies against caspase-1 (p48 and p20 fraction) (1:2000) (#Cat. AG-20B-0042; AdipoGen Life Science, Liestal, Switzerland) and p-p65 NF-κB (1:100) (#Cat. 10745-1-AP; Proteintech^®^ Group, Rosemont, IL, USA) overnight at 4 °C. Finally, membranes were incubated with peroxidase-conjugated polyclonal goat anti-rabbit immunoglobulin (Ig)G (1:5000) (#Cat. P0448; Dako, Glostrup, Denmark) for p-p65 NF-κB and peroxidase-conjugated polyclonal goat anti-mouse IgG (1:4000) (#Cat. A4416; Sigma-Aldrich, St. Louis, MO, USA) for caspase-1 for 1 h at room temperature. β-actin (1:500) (#Cat. A2066; Sigma-Aldrich, St. Louis, MO, USA) or glyceraldehyde-3-phosphate dehydrogenase (GADPH) (1:2500) (#Cat. G9545; Sigma-Aldrich, St. Louis, MO, USA) was used as the protein loading control. The immunoreactive bands were visualized through enhanced chemiluminescence (ECL, RPN2232; Amersham, GE Healthcare Life Sciences, Barcelona, Spain) using an AutoChemi System (P/N97-0150-02) imager and LabWorks version 4.6 image acquisition software (UVP Inc., Upland, CA, USA). Band intensity was assessed through optical densitometry with the use of the Fiji downloads (Windows 32-bit) platform, an image-processing package distribution of ImageJ2 [40]. In brief, after uploading the image, the type 32-bit option was selected in the Image submenu. A horizontal rectangular selection tool was used to outline all the bands of the protein of interest. Then, the commands ‘First Lane’ and ‘Plot lanes’ from the submenu to analyse one-dimensional electrophoretic gels, followed by the ‘Wand’ tool, were used to determine the optical density of each peak area. This process was repeated for each protein of interest and the internal loading controls β-actin and GAPDH. Data were exported to an Excel book, and the intensity protein of interest band/intensity of internal loading band ratio was calculated.

### 4.6. Determination of Mitochondrial ROS with MitoSOX^TM^

Elicited macrophages were plated on an eight-well Lab-Tek chamber slide (Nunc –Thermo Fisher, Rochester, NY, USA) at 0.5 × 10^6^ cells/well in 500 µL. After adherence, cells were incubated with osteostatin in the absence or presence of LPS (1 μg/mL) + ATP (5 mM) stimulus. The medium was removed, and 5 µM of MitoSOX^TM^ (#Cat. M36008; Molecular ProbesTM, Invitrogen, Paisley, UK) was added in 500 µL of Hank’s salt solution (HBSS) with Ca^2+^ and Mg^2+^. After incubation at 37 °C for 10 min, cells were washed with PBS and fixed with the use of 4% (wt/vol) p-formaldehyde for 15 min. After several washes, cells were stained with ProLong^TM^ Gold Antifade Mountant with DAPI (#Cat. P36935; Molecular Probes TM Invitrogen, Paisley, UK). Six fields per well were examined under a confocal microscope Olympus FV1000 (Waltham, MA, USA).

### 4.7. Determination of Extracellular ROS by Chemiluminescence

Peritoneal macrophages were cultured at 10^6^ cells/mL in 96-well plates. After cell adherence, the medium was replaced by 200 µL of HBSS with Ca^2+^ and Mg^2+^. Cells were treated with osteostatin for 30 min and then stimulated with 12-O-tetradecanoylphorbol-13-acetate (TPA) (#Cat. P8139; Sigma-Aldrich^®^, St. Louis, MO, USA) (10^−6^ M) for 20 min in the presence of luminol (#Cat. A8511; Sigma-Aldrich^®^, St. Louis, MO, USA) (40 µM). The chemiluminescence produced was measured using the Wallac 1420 VICTOR3^TM^ microplate reader (PerkinElmer, Turku Finland).

### 4.8. Determination of Nrf2 Translocation in the Nucleus by Immunofluorescence

Elicited peritoneal macrophages were plated on an eight-well Lab-Tek chamber slide (Nunc–Thermo Fisher, Rochester, NY, USA) at 0.5 × 10^6^ cells/well in 500 µL. After adherence, cells were incubated with osteostatin in the presence or absence of LPS (1 μg/mL) + ATP (5 mM) stimulus. Then, cells were washed with PBS and fixed with methanol at 4 °C for 15 min. After washing with PBS, cells were incubated with anti-Nrf2 (#Cat. ab137550; Abcam, Cambridge, UK) at 4 °C for 18 h and then with Alexa Fluor^®^ 488 goat anti-rabbit (#Cat. A11008; Molecular ProbesTM Invitrogen, Paisley, UK) for 1 h at room temperature. Cells were stained with ProLongTM Gold Antifade Mountant with DAPI (#Cat. P36935; Molecular ProbesTM Invitrogen, Paisley, UK) and fixed for 18 h before visualization under a confocal microscope Olympus FV1000 (Waltham, MA, USA).

### 4.9. Mouse Air Pouch Model Induced by Calcium Pyrophosphate Dihydrate (CPPD) Crystals

To create the air pouch, 10 mL of sterile air was injected subcutaneously into the dorsal area of mice (day 0). Three days later, 5 mL of sterile air was injected (day 3), and 6 days after the initial injection of sterile air (day 6), mice were treated via the intra-pouch administration of osteostatin dissolved in 100 µL of saline solution at 3 µg/pouch or 6 µg/pouch. After 20 min, 1 mL of calcium pyrophosphate dihydrate crystals (CPPD) (#Cat. tlrl-cppd, InvivoGen, San Diego, CA, USA) (1 mg/mL in sterile PBS) was injected into the pouches of the control and treated groups, and 1 mL of sterile PBS was injected into the naïve group [22]. After 6 h, mice were anesthetized and then euthanized via cervical dislocation, and air pouch exudates were collected. Cells present in exudates were counted with a Coulter counter (Beckman Coulter™ Z2 Coulter^®^, Indianapolis, IN, USA). Exudates were centrifuged at 1200× *g* for 10 min at 4 °C. Supernatants were then collected and used to determine the levels of the cytokines IL-1β, IL-18, TNFα, CXCL-1, and IL-6 (ELISA); the expression of p20 caspase-1 (Western blotting); and myeloperoxidase activity. Cell pellets were lysed and used for p48 caspase-1 and p-p65 NF-κB determination using Western blotting.

### 4.10. Myeloperoxidase Activity Determination

Supernatants from air pouch exudates were incubated with PBS (pH 7.4) and phosphate buffer pH 5.4 (Na_2_HPO_4_ 0.09%, NaH_2_PO_4_ 1.15%) in the presence of hydrogen peroxide (0.05%) for 5 min. Next, TMB at 18 mM dissolved in dimethylformamide (prepared at 8% in distilled water) was added. After 3 min of incubation at 37 °C, the reaction was stopped with 2 N sulphuric acid. The absorbance was quantified using a Wallac 1420 VICTOR3^TM^ spectrophotometer (PerkinElmer, Turku, Finland) at 450 nm [41].

### 4.11. Mouse Model of Gouty Arthritis Induced by Monosodium Urate (MSU) Crystals

Osteostatin dissolved in saline solution (100 µL) at 80 µg/kg or 120 µg/kg was administered subcutaneously on the dorsum of mice. One hour later, 2 mg of MSU crystals (#Cat. tlrl-msu-25; InvivoGen; San Diego, CA, USA) resuspended in 50 μL of sterile PBS was injected subcutaneously into the plantar aponeurosis of the right hind paw [23]. Then, 1 h, 3 h, 6 h, and 24 h after the injection of the MSU crystals, the edema was measured using a digital plethysmometer (Digital Water Plethysmometer, Panlab S.L.U., Barcelona, Spain). After 24 h, the animals were anesthetized and then euthanized via cervical dislocation, and limbs were surgically removed (by scissors) and frozen at −80 °C for subsequent homogenization in liquid N_2_ for the measurement of inflammatory mediators and Western blotting.

### 4.12. Determination of Mediators in Paw Homogenates

Hind limbs were homogenized in liquid N_2_ with 1 mL of buffer A, pH 7.4 (10 mM HEPES, pH 8 1 mM EDTA, 1 mM EGTA, 10 mM KCl, 1 mM dithiothreitol, 5 mM NaF, 1 mM Na_3_VO_4_, 10 mM Na_2_MoO_4_, 1 mg/mL leupeptin, 0.1 mg/mL aprotinin, and 0.5 mM phenylmethylsulfonyl fluoride). Tissue homogenates were sonicated (3 × 10 s) on ice, incubated for 10 min at 4 °C, and centrifuged at 1500× *g* for 5 min at 4 °C. Supernatants were collected and centrifuged at 10,000× *g*, for 5 min at 4 °C. Finally, supernatants were used for the determination of cytokines (IL-1β, TNF-α, IL-18, IL-6, CXCL-1) via ELISA, while MPO activity and protein expression (caspase-1 (p20 and p48) and p-65 NF-κB) were determined via Western blotting.

### 4.13. Statistical Analysis

Data were analyzed using GraphPad Prism 7.0 (GraphPad Software Inc., La Jolla, CA, USA). All values are expressed as mean ± standard deviation (SD). Statistical significance was determined via one-way analysis of variance (ANOVA) with a post hoc Tukey’s test for multiple group comparisons, and every possible comparison between the study groups was considered. Alternatively, two-way ANOVA analysis was performed for two independent variables, time and treatment vs. edema volume, with a post hoc Bonferroni test used for multiple group comparisons considering the repetition at different times in the MSU crystals model. Results with *p* < 0.05 were considered statistically significant.

## Figures and Tables

**Figure 1 ijms-25-02752-f001:**
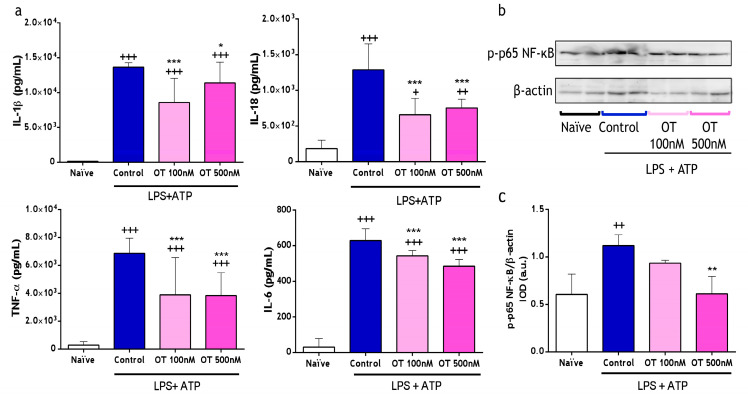
Levels of secreted cytokines and NF-κB activation by peritoneal macrophages stimulated with lipopolysaccharide (LPS) and adenosine triphosphate (ATP). (**a**) Cytokine levels in cell supernatants were measured via ELISA (6 independent experiments in duplicate). (**b**) A representative image of Western blotting analysis of p-p65-NF-κB expression on protein extracts from cell lysates of peritoneal macrophages. β-actin is the loading control. (**c**) p-p65-NF-κB/β-actin is expressed as arbitrary units (a.u.) of integrated optical density (IOD) (4 independent experiments). Values are presented as mean ± S.D. + *p* < 0.05, ++ *p* < 0.01, and +++ *p* < 0.001 versus naïve group; * *p* < 0.05, ** *p* < 0.01, and *** *p* < 0.001 versus control group. One-way ANOVA with Tukey’s post-test in (**a**,**c**) to measure statistical significance. OT: osteostatin; LPS: lipopolysaccharide; ATP: adenosine triphosphate.

**Figure 2 ijms-25-02752-f002:**
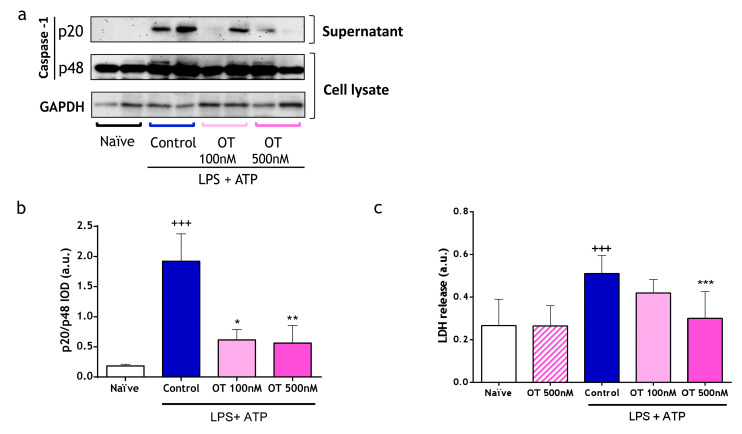
Caspase-1 expression and lactate dehydrogenase (LDH) release on peritoneal macrophages stimulated with LPS and ATP. (**a**) Representative image of Western blotting analysis of p20 and p48 subunits of caspase-1 expression on protein extracts from supernatants and cell lysates of peritoneal macrophages. (**b**) p20/p48 is expressed as arbitrary units (a.u.) of integrated optical density (IOD) (5 independent experiments). (**c**) LDH release was measured using a colorimetric assay (6 independent experiments). Data presented as mean ± S.D. +++ *p* < 0.001 versus naïve group; * *p* < 0.05, ** *p* < 0.01, and *** *p* < 0.001 versus control group. One-way ANOVA (Tukey’s post-test) in (**a**,**c**). OT: osteostatin.; LPS: lipopolysaccharide; ATP: adenosine triphosphate; LDH: lactate dehydrogenase.

**Figure 3 ijms-25-02752-f003:**
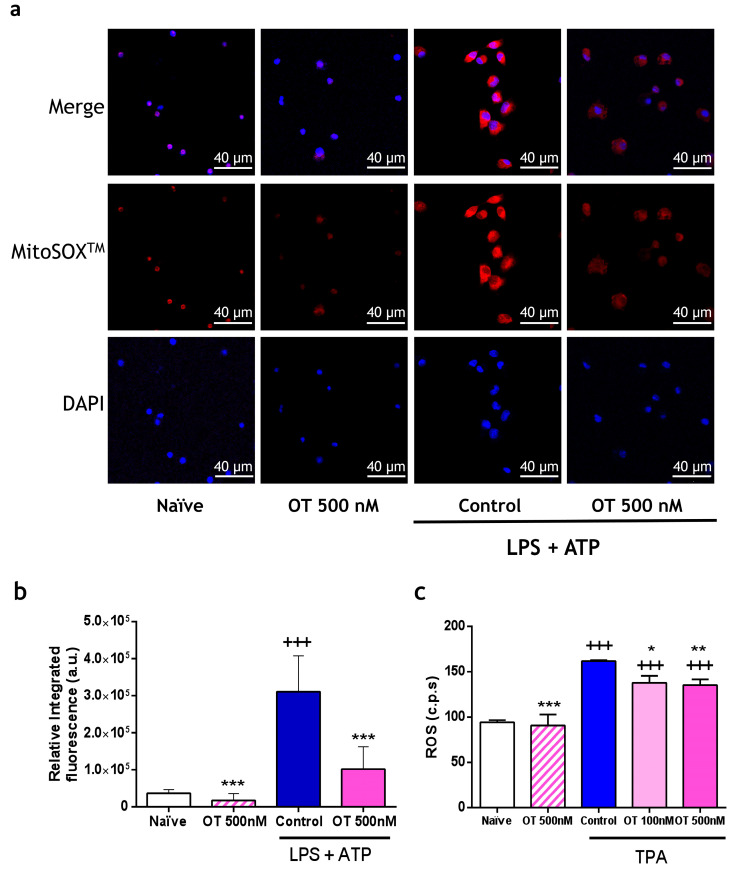
Levels of mitochondrial (**a**,**b**) and extracellular (**c**) reactive oxygen species (ROS) production by peritoneal macrophages. (**a**) Representative image of the fluorescence obtained via reaction with MitoSOX^MT^ using an Olympus FV1000 confocal microscope; cell nuclei were counterstained in blue (DAPI). Scale bar = 40 μm. (**b**) Integrated intensity obtained from the ratio of mean fluorescence emitted via MitoSOX^MT^ to cell area (4 independent experiments). (**c**) Extracellular ROS were measured via chemiluminescence in counted photons per second (cps) (6 independent experiments). Data presented as mean ± S.D. +++ *p* < 0.001 versus naïve group; * *p* < 0.05, ** *p* < 0.01, and *** *p* < 0.001 versus control group. One-way ANOVA (Tukey’s post-test) in (**b**,**c**). OT: osteostatin; LPS: lipopolysaccharide; ATP: adenosine triphosphate; TPA: 12-O-tetradecanoylphorbol-13-acetate.

**Figure 4 ijms-25-02752-f004:**
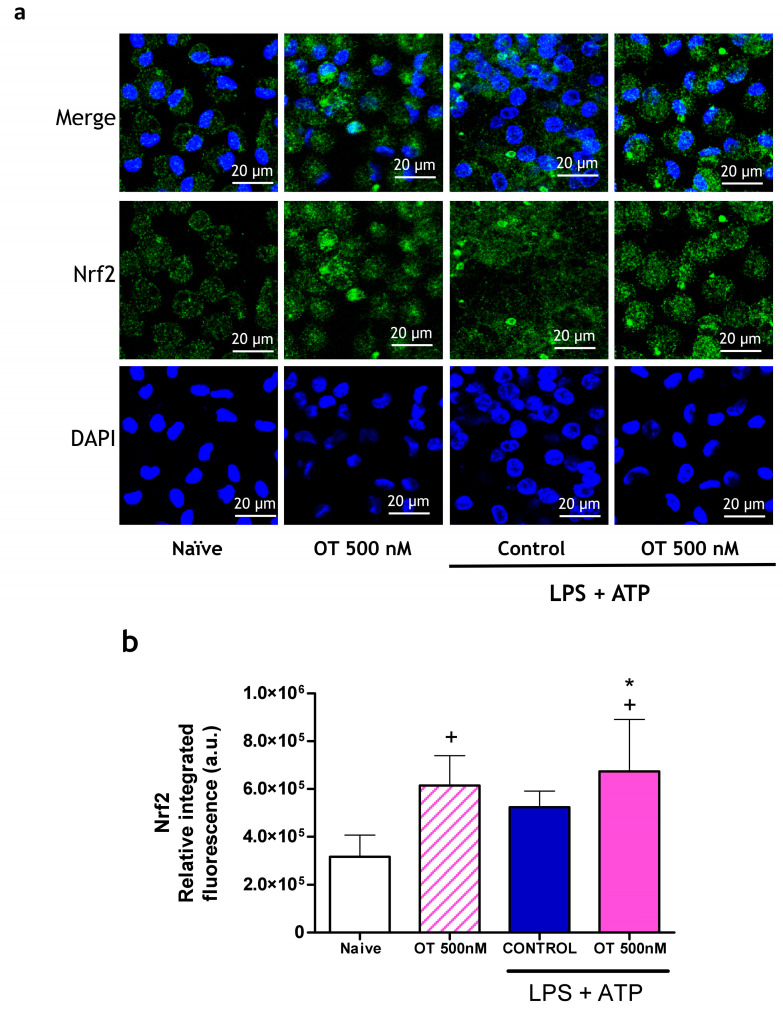
Nrf2 expression by peritoneal macrophages. Nrf2 nuclear translocation was determined via immunofluorescence of peritoneal macrophages stimulated with LPS + ATP in the presence or absence of osteostatin. (**a**) Representative confocal images of macrophages showing Nrf2 transcription to the nuclei (labeled in green). Cell nuclei were counterstained in blue (DAPI). Scale bar = 20 μm. (**b**) Integrated intensity obtained from ratio of mean fluorescence to cell area. Data presented as mean ± S.D. of 4 independent experiments. + *p* < 0.05 versus naïve group; * *p* < 0.05 versus control group. One-way ANOVA (Tukey’s post-test). OT: osteostatin; LPS: lipopolysaccharide; ATP: adenosine triphosphate.

**Figure 5 ijms-25-02752-f005:**
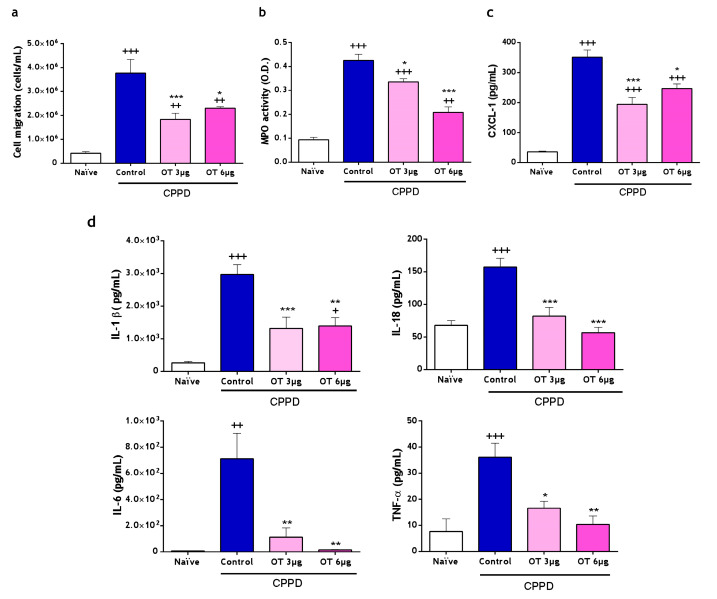
Cell migration (**a**), myeloperoxidase activity, (**b**) and pro-inflammatory cytokine (**c**,**d**) release in mouse air pouch (MAP) exudates. (**a**) Cell numbers in MAP exudates were measured with the use of a Coulter counter. (**b**) Myeloperoxidase (MPO) activity was determined via spectrophotometry O.D. (optical density, 450 nm). (**c**,**d**) Cytokines were measured via ELISA. Data were presented as mean ± S.D. *n* refers to the number of animals used in each group. *n* = 6 in the naïve group, and *n* = 8 in each of the following groups (the control, OT3µg and OT6µg groups). + *p <* 0.05, ++ *p* < 0.01, and +++ *p* < 0.001 versus the naïve group; * *p* < 0.05, ** *p* < 0.01, and *** *p* < 0.001 versus the control group. One-way ANOVA (Tukey’s post-test). OT: osteostatin; CPPD: calcium pyrophosphate dihydrate.

**Figure 6 ijms-25-02752-f006:**
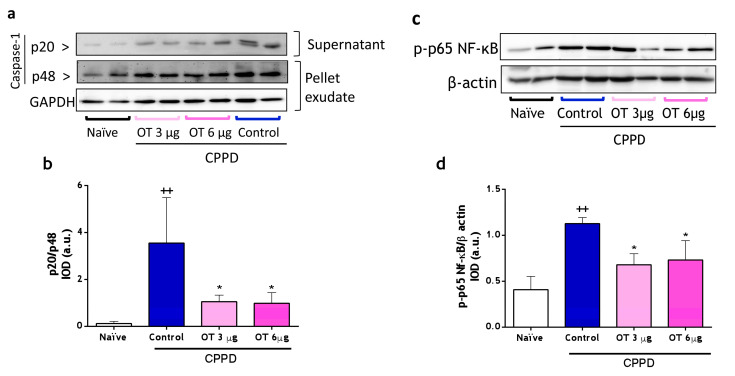
Caspase-1 and NF-κB expression in mouse air pouch (MAP) exudates. (**a**) Western blotting analysis of p20 and p48 subunits of caspase-1 expression on protein extracts from supernatants and cell lysates of MAP exudates. (**b**) p20/p48 is expressed as arbitrary units (a.u.) of integrated optical density (IOD). (**c**) Representative Western blotting analysis of p-p65-NF-κB expression in cell lysates of MAP exudates. (**d**) p-p65-NFkB/β-actin is expressed as arbitrary units (a.u.) of integrated optical density (IOD). Data presented as mean ± S.D. *n* = 4 in the naïve group and *n* = 5 in each of the following groups. ++ *p* < 0.01 versus the naïve group; * *p* < 0.05 versus the control group. One-way ANOVA (Tukey’s post-test). OT: osteostatin; CPPD: calcium pyrophosphate dihydrate crystals.

**Figure 7 ijms-25-02752-f007:**
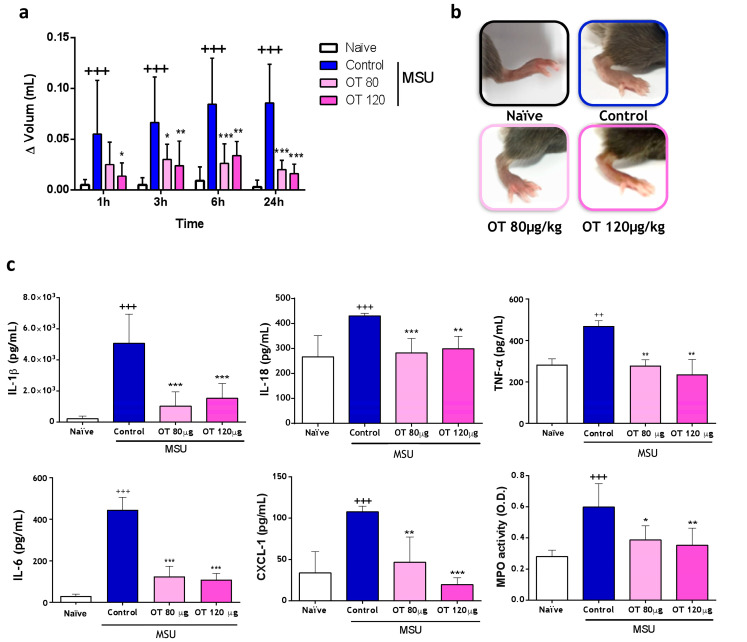
Edema in monosodium urate (MSU) crystal-induced gouty arthritis and cytokines levels and MPO activity in paw homogenates. (**a**) Assessment of edema induced by MSU crystals at different time points (1 h, 3 h, 6 h, and 24 h after MSU administration) with a digital plethysmometer. (**b**) Representative images of mice right hind paw at 24 h after MSU crystal administration. (**c**) Cytokines were determined via ELISA. Myeloperoxidase (MPO) activity was measured via spectrophotometry O.D. (optical density, 450 nm). Data presented as mean ± S.D. *n* = 6 in the naïve group and *n* = 8 in each of the following groups. +++ *p* < 0.001; ++ *p* < 0.01 versus the naïve group; * *p* < 0.05, ** *p* < 0.01, and *** *p* < 0.001 versus the control group. Two-way ANOVA with the Bonferroni post-test in (**a**) and one-way ANOVA with Tukey’s post-test in (**c**). OT: osteostatin.; MSU: monosodium urate crystals; MPO: myeloperoxidase.

**Figure 8 ijms-25-02752-f008:**
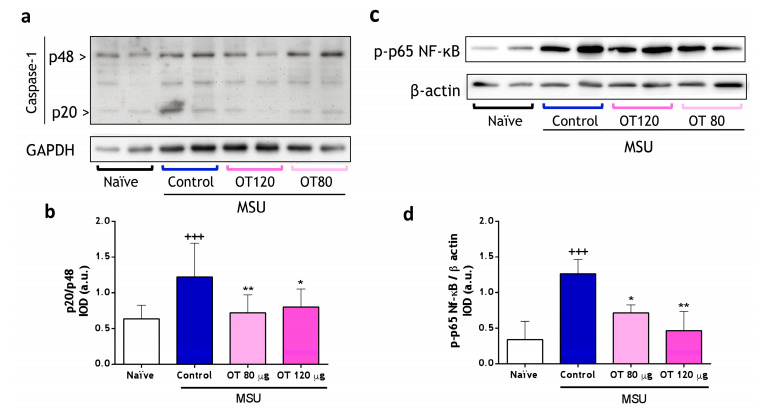
Effect of osteostatin on caspase-1 expression and p65 NF-κB phosphorylation in paw homogenates in monosodium urate (MSU) crystal-induced gouty arthritis. (**a**) Representative image of Western blotting analysis of p20 and p48 subunits of caspase-1 expression in protein extracts from paw homogenates. (**b**) p20/p48 is expressed as arbitrary units (a.u.) of integrated optical density (IOD). (**c**) Representative image of Western blotting analysis of p-p65-NF-κB expression in paw homogenates. (**d**) p-p65-NFkB/β-actin is expressed as arbitrary units (a.u.) of integrated optical density (IOD). Data presented as mean ± S.D. *n* = 4 in the naïve group and *n* = 5 in each of the following groups. +++ *p* < 0.001 versus the naïve group; * *p* < 0.05 and ** *p* < 0.01 versus the control group. One-way ANOVA (Tukey’s post-test). OT: osteostatin; MSU: monosodium urate crystals.

## Data Availability

All relevant data are presented in the manuscript. Raw data are available upon request from the corresponding authors.

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
