# Peer review of "Osteostatin Mitigates Gouty Arthritis through the Inhibition of Caspase-1 Activation and Upregulation of Nrf2 Expression"

_ijms, 2024, doi:10.3390/ijms25052752_

Round 1
Reviewer 1 Report
Comments and Suggestions for Authors
Paper titled (Osteostatin mitigates gouty arthritis through the inhibition of caspase-1 activation and upregulation of Nrf2 expression.) by Catalan et al. studied the impact of using osteostatin on gouty arthritis model and claimed the improving effect was attributed to downregulation of caspase 1 activity and upregulation in Nrf2 expression. The aim here is straight forward but the study is not of high clinical potential. The main concern in writing this paper is the METHODS which lacks important refernces. I have some recommendations for improving the manuscript.
1- Abstract: should be amended by some numerical values from the results
2- Key words are above the limit & some of them are not necessary as "crystals" and cytokines.
Further the animal species should be mentioned
3- Introduction: mention the current therapeutic options for gout & their limitations
4- Explain more the rational why the authors thought that osteostatin may improve gouty arthritis, this is not clear
5- Revise punctuation in general and in Introduction.
6- In the aim, mention how you achieved it and measurements done
7- Results: Figure legends: no need to repeat that data are mean and SE several times in the same figure . one time is enough
8- Data better be presented as mean+-SD (not SE) this is as authors do not cover the universe for this study.
9- Mention "n" in each illustation individually
10- No need to mention P <0.001 or P<0.01; as P<0.05 is enough and high p levels does not mean the mean values are greatly different; just tell that the SD values are small.
11- Every abbreviation in figures should be explained in the figure legend to be self explanatory & stands alone.
12- In methods: Authors should confirm in methods that "every possible comparison between the study groups was considered" and apply this in results.
13- Why some figures include 5 groups & some includes 4 groups? this is inconsistent and does not imply a good fair comparison and conclusion
14- Methods: How euthanasia was performed and tissues dissected?
15- Detailed housing conditions and feeding should be provided & how authors minmized animal suffering
16 - Give the method of solubilizing or dilution of the drugs.
17- Authors have to check the normality of distribution of the results by a suitable post hoc test (such as Shapiro-Wilk test or K-S test) before deciding to choose certain ANOVA. If the normality test indicated normal dist of the data, so use one-way or 2 -way ANOVA, if not, use non parametric ANOVA tests. In all cases choose a suitable post-hoc test
18- Give brief account on the type of ELISA assays
18- Methods in general lacks references!!! at many occasions. Please add the apparopriate refernces in each step
19- The method and software used for quantification in WB analysis should be added.
20- Use appropriate abbreviations for minutes, seconds...etc
21-Authors should give the source of chemicals, kits and antibodies completely and consistently (code, company, town, state and country) & version for software
22- Authors did not provide the rational why decided to use one-way ANOVA in some cases & 2 way ANOVA in some cases
Comments on the Quality of English Languagefine
Author Response
1. Abstract: should be amended by some numerical values from the results
RESPONSE: Due to the abstract word limit we did not follow this suggestion.
2- Key words are above the limit & some of them are not necessary as "crystals" and cytokines.
RESPONSE: We have deleted both terms as keywords.
3. Introduction: mention the current therapeutic options for gout & their limitations.
RESPONSE: We have included a paragraph (lines 53-58). Pharmacotherapeutic treatment of gouty arthritis from two points of view: symptomatic or anti-inflammatory treatment and treatment of hyperuricemia. In this case, we focused on the anti-inflammatory treatment. The current anti-inflammatory treatment includes colchicine, non-steroidal anti-inflammatory drugs (NSAIDs), steroids and IL-1 blockers).
4. Explain more the rational why the authors thought that osteostatin may improve gouty arthritis, this is not clear.
RESPONSE: We have amended the aims of the study in the last paragraph of the introduction (lines 81-83) to better state the interest of studying the effect of osteostatin as a potential candidate for acute gouty arthritis.
5. Revise punctuation in general and in Introduction.
RESPONSE: We have revised the punctuation.
6- In the aim, mention how you achieved it and measurements done.
RESPONSE: We have mentioned the aims it in lines 83-87.
7- Results: Figure legends: no need to repeat that data are mean and SE several times in the same figure. one time is enough.
RESPONSE: We have already modified the text of all figure legends according to the recommendations.
8- Data better be presented as mean+-SD (not SE) this is as authors do not cover the universe for this study.
RESPONSE: We thank the reviewer for this observation and have proceeded accordingly.
9- Mention "n" in each illustration individually
RESPONSE: We have modified all figure legends to better describe the n in each experiment and group.
10- No need to mention P <0.001 or P<0.01; as P<0.05 is enough and high p levels does not mean the mean values are greatly different; just tell that the SD values are small.
RESPONSE: We have preferred to keep the statistical significance as per the software used for the data analysis.
11- Every abbreviation in figures should be explained in the figure legend to be self-explanatory & stands alone.
RESPONSE: We have stand-alone and explain the abbreviations CPPD, MSU, ATP, LDH, TPA, and LPS in each figure.
12- In methods: Authors should confirm in methods that "every possible comparison between the study groups was considered" and apply this in results.
RESPONSE: We have added this statement accordingly in the section 4.13 Statistical analysis (lines 603-604) of the Methods section.
13- Why some figures include 5 groups & some includes 4 groups? This is inconsistent and does not imply a good fair comparison and conclusion.
RESPONSE: In some protocols we have decided it was interesting to assay OT500nM in the absence of stimuli in order to rule out its toxicity (Figure 2b) or its pro-oxidant effect (figure 3c). To point out these observations we have added two statements in the corresponding sections (lines 124-125 and 146-148) On the other hand, in the fluorescent studies with macrophages the choice of the highest OT dose was due to the limitation of the experimental setting with an eight-well chamber.
14- Methods: How was euthanasia performed and tissues dissected?
RESPONSE: In the mouse model of gouty arthritis induced by monosodium urate (MSU) crystals, mice were euthanized by cervical dislocation as previously stated in the previous model. We have added this statement in line 585. And added that leg dissection was performed by scissors (line 586).
15- Detailed housing conditions and feeding should be provided & how authors minimized animal suffering
RESPONSE: In response to this suggestion, we have provided detailed information on the animals used and housing conditions and welfare monitoring in the Materials and Methods section in 4.1. Animals (lines 432-446). Specifically, we used male C57BL/6 mice sourced from Charles River, Écully, France, aged between 10 and 12 weeks, with a weight range of 20-25 g. The text states like this: Male C57BL/6 mice (Charles River, Écully, France) between 10 and 12 weeks of age (20-25 g).
16 - Give the method of solubilizing or dilution of the drugs.
RESPONSE: The peptide osteostatin was osteostatin (1-5) amide (human, bovine, dog, horse, mouse, rabbit, rat) trifluoroacetate salt (Ref: 4025761.0025; Bachem, Bubendorf, Switzerland) was dissolved in a saline solution. We have added the information of osteostatin and the method of dilution of the drug in red in Material and Methods section in 4.2. Isolation and culture of peritoneal macrophages. The information added is as follows: “Osteostatin (1-5) amide trifluoroacetate salt (Ref: 4025761.0025; Bachem, Bubendorf, Switzerland) was first dissolved in saline solution to achieve a concentration of 10µM from which subsequent dilutions of 100 nM and 500 nM were prepared in the culture medium to conduct macrophage studies”; in 4.9. Mouse air pouch model induced by calcium pyrophosphate dihydrate (CPPD) crystals, the added information is “mice were treated with intra-pouch administration of osteostatin dissolved in 100 µL of saline solution at 3 µg or 6 µg per pouch”; in 4.11. Mouse model of gouty arthritis induced by monosodium urate (MSU) crystals, the added information is “Osteostatin dissolved in saline solution (100 µL) at 80 µg/kg or 120 µg/kg was administered subcutaneously on the dorsum of mice.”
17- Authors have to check the normality of distribution of the results by a suitable post hoc test (such as Shapiro-Wilk test or K-S test) before deciding to choose certain ANOVA. If the normality test indicated normal dist. of the data, so use one-way or 2 -way ANOVA, if not, use nonparametric ANOVA tests. In all cases choose a suitable post-hoc test.
RESPONSE: We have obtained normal distribution; therefore, one-way or two-way ANOVA was used.
18- Give brief account on the type of ELISA assays
RESPONSE: all ELISA performed in this study were sandwich ELISA. It is mentioned in section 4.4. ELISA in Material and methods section (line 486).
19- Methods in general lacks references!!! at many occasions. Please add the appropriate references in each step.
RESPONSE: We have added all catalogue numbers and references in methods section (whole methods section).
20- The method and software used for quantification in WB analysis should be added.
RESPONSE: The quantification of the Western Blot images was performed using ImageJ software, version 1.52 (line 516).
21- Use appropriate abbreviations for minutes, seconds...etc.
RESPONSE: we have already modified the abbreviations accordingly.
22-Authors should give the source of chemicals, kits and antibodies completely and consistently (code, company, town, state and country) & version for software
RESPONSE: As stated in answer 18 and 19, catalogue numbers, references and software version were added.
23- Authors did not provide the rational why decided to use one-way ANOVA in some cases & 2-way ANOVA in some cases.
RESPONSE: We have specified the criteria in section 4.13 of the methods (lines 600-607).
Reviewer 2 Report
Comments and Suggestions for Authors
The research paper on osteostatin and gouty arthritis presents some interesting findings The review aims to Investigate the possible links between two prevalent age-related diseases – OA and AD – that hold significant research potential. Gouty arthritis is a common and debilitating disease, and finding new therapeutic options is crucial. The study focuses on the potential of osteostatin as a treatment for gouty arthritis, investigating its effects on several key pathways. The research employs both in vitro and in vivo models, providing a strong foundation for the findings. The results demonstrate a clear link between osteostatin administration and reduced inflammation, pyroptosis, and oxidative stress in both cell and animal models.
There are some critical comments and suggestions for improvement:
· The introduction does not provide a clear rationale for the study or a clear research question. It would be helpful to explain why osteostatin is a potential candidate for treating gouty arthritis, what are the gaps in the current literature, and what are the specific aims and hypotheses of the study.
· While the anti-inflammatory properties of osteostatin are promising, highlight how this study specifically advances the understanding of its role in gouty arthritis compared to existing research.
· Although the study identifies key pathways, delve deeper into the precise mechanisms by which osteostatin might influence caspase-1, NF-κB, and Nrf2. Consider involving specific protein interactions or signaling molecules.
· Acknowledge potential limitations of the study, such as the use of specific cell lines or animal models, and discuss future directions for investigation.
· Ensure clarity and conciseness in presenting data through figures and tables. Consider statistical analysis and significance testing where appropriate.
· The methods section is not detailed enough to allow replication of the study. It would be useful to provide more information on the sources and characteristics of the MSU and CPPD crystals, the doses and routes of administration of osteostatin, the time points and methods of sample collection, and the statistical analyses performed.
· The results section does not present the data clearly and consistently. It would be better to use tables and figures to summarize the main results, and to avoid repeating the same information in the text. It would also be important to report the effect sizes and confidence intervals of the comparisons and to perform appropriate tests for multiple comparisons.
· Explore the potential side effects or off-target effects of osteostatin, especially in the context of long-term administration.
· Discuss the feasibility and potential challenges of translating these findings into clinical applications for gouty arthritis patients.
· Consider comparing the efficacy of osteostatin with existing treatment options for gouty arthritis.
· The discussion section does not interpret the results in light of the existing literature or discuss the implications and limitations of the study. It would be advisable to compare and contrast the findings with previous studies on osteostatin and gouty arthritis, to explain the possible mechanisms of action of osteostatin, to acknowledge the potential confounding factors and sources of bias, and to suggest directions for future research.
· The conclusion should succinctly summarize the key findings and their implications for gouty arthritis treatment. Consider briefly mentioning future research directions.
By addressing these suggestions and further strengthening the analysis and interpretation of your findings, you can elevate your research to a high level of impact and contribute significantly to the development of novel therapeutic strategies for gouty arthritis.
Author Response
1- The introduction does not provide a clear rationale for the study or a clear research question. It would be helpful to explain why osteostatin is a potential candidate for treating gouty arthritis, what are the gaps in the current literature, and what are the specific aims and hypotheses of the study.
RESPONSE: We have amended the introduction to better describe the rationale of the study and the potential interest given the current therapeutic options available (lines 53-58 and 81-87).
2- While the anti-inflammatory properties of osteostatin are promising, highlight how this study specifically advances the understanding of its role in gouty arthritis compared to existing research.
RESPONSE: We acknowledge the interest of this comment, and have included a small discussion about the promising benefits of osteostatin in comparison with the actual anti-inflammatory therapeutic options in gouty arthritis (lines 409-420). In addition, we have highlighted in the introduction that prior research with osteostatin focused more on its effect on bone metabolism and rheumatoid arthritis, but little is known about its effect on gouty arthritis.
3- Although the study identifies key pathways, delve deeper into the precise mechanisms by which osteostatin might influence caspase-1, NF-κB, and Nrf2. Consider involving specific protein interactions or signaling molecules.
RESPONSE: Absolutely, this is a very interesting topic that we have planned to delve deeper in future actions. As this is the first time, to our knowledge, that it has been investigated the influence of osteostatin on these pathways and its benefits on two animal models of gouty arthritis, more detailed mechanistic studies will be performed in the future.
4- Acknowledge potential limitations of the study, such as the use of specific cell lines or animal models and discuss future directions for investigation.
RESPONSE: As mentioned in the previous comment, future investigations will be performed to study the mechanism involved in caspase-1, NF-κB, and Nrf2 pathways using human macrophages isolated from whole peripheral blood. As one of the main limitations is that our data are obtained from animal primary cells and experimental models that are not always translatable to human diseases (lines 409-420).
5- Ensure clarity and conciseness in presenting data through figures and tables. Consider statistical analysis and significance testing where appropriate.
RESPONSE: We have reanalyzed the data and their graphical representation. In addition, we have clarified in methods section, 4.13 Statistical Analysis, the criteria for using one- or two- way statistical data analysis.
6- The methods section is not detailed enough to allow replication of the study. It would be useful to provide more information on the sources and characteristics of the MSU and CPPD crystals, the doses and routes of administration of osteostatin, the time points and methods of sample collection, and the statistical analyses performed.
RESPONSE: Further details, references and catalogues numbers have been added in the methods section to ensure the replicability of the studies.
7- The results section does not present the data clearly and consistently. It would be better to use tables and figures to summarize the main results, and to avoid repeating the same information in the text. It would also be important to report the effect sizes and confidence intervals of the comparisons and to perform appropriate tests for multiple comparisons.
RESPONSE: Although we cannot deny the interest of this comment, we have followed the graphical description and data analysis recommended by the statistician consultant of the Ethics Committee for the Welfare of Experimentation animals of the University of Valencia. According to him, the confidence intervals are for frequency analysis with higher numbers of individuals. On the other hand, all statistical comparisons have been performed among all the study groups. We have added a statement in the methods section (lines 603-604).
8- Explore the potential side effects or off-target effects of osteostatin, especially in the context of long-term administration.
RESPONSE: We have focused on the acute attack just to highlight the efficacy of osteostatin. Osteostatin is a novel substance, which is being studied in different pathological conditions. Surely, for longer-term studies safety studies should be undertaken. On this regard, we have expanded in the discussion section (lines 409-416) that, after 15 daily-administration of s.c. osteostatin (120μg/kg), there was no visible sign of toxicity or behavioral change in the chronic murine arthritis model induced by Collagen II (doi: 10.3390/ijms20163845).
9- Discuss the feasibility and potential challenges of translating these findings into clinical applications for gouty arthritis patients.
RESPONSE: In fact, our results are quite preliminary to discuss the translatability in the short future, for which administration routes and doses will be the most appropriate in clinical studies. In the present study, we have observed the potential benefit of this peptide administered s.c. on the back of mice, far from the site of induction of gouty arthritis by MSU crystals. We have further discussed the advantages for drug development of this pentapeptide, with less immunogenicity and more favorable pharmacokinetic, in comparison to other higher MW peptide products (lines 409-420).
10- Consider comparing the efficacy of osteostatin with existing treatment options for gouty arthritis.
RESPONSE: Despite the interest of the comparison with current anti-inflammatory therapeutic options, the main objective of our study was to prove its efficacy, given the interesting results observed in vitro. We have added in the discussion, as in comment 8. The potential advantages.
11- The discussion section does not interpret the results in light of the existing literature or discuss the implications and limitations of the study. It would be advisable to compare and contrast the findings with previous studies on osteostatin and gouty arthritis, to explain the possible mechanisms of action of osteostatin, to acknowledge the potential confounding factors and sources of bias, and to suggest directions for future research.
RESPONSE: To our knowledge this is the first report addressing the beneficial effect of osteostatin on gouty arthritis. Prior reports study its effects on other experimental models already discussed in the manuscript. As mentioned earlier, we have added in the discussion the limitations and future research directions.
12- The conclusion should succinctly summarize the key findings and their implications for gouty arthritis treatment. Consider briefly mentioning future research directions.
RESPONSE: We have amended the conclusion accordingly (lines 422-425).
By addressing these suggestions and further strengthening the analysis and interpretation of your findings, you can elevate your research to a high level of impact and contribute significantly to the development of novel therapeutic strategies for gouty arthritis.
RESPONSE: We thank the reviewer for all the suggestions, and agree that by answering them the quality of the manuscript had greatly improved.
Round 2
Reviewer 1 Report
Comments and Suggestions for Authors
The revised version of paper titled (Osteostatin mitigates gouty arthritis through the inhibition of caspase-1 activation and upregulation of Nrf2 expression.) is improved but 2 critical points should be assessed: Otherwise, the paper can never be accepted
- Authors mentioned that "In the mouse model of gouty arthritis induced by monosodium urate (MSU) crystals, mice were euthanized by cervical dislocation"
So authors used cervical dislocation technique without anesthesia? this is not permittable. How the ethical committee approved this protocol?
- The method and software used for quantification in WB analysis are still not enough to allow reproducibility of the methods
- Results should be consistent in number of groups. Neglection of one group due to any reason is not accepted
Authors mentioned they applied every possible comparison in methds. OK, but did you already applied the criteria in assessing the results? as no changes were made in figures.
Comments on the Quality of English Language
fine
Author Response
1- Authors mentioned that "In the mouse model of gouty arthritis induced by monosodium urate (MSU) crystals, mice were euthanized by cervical dislocation"
RESPONSE: We have corrected the statements in both animal models by adding that animals were anesthetized with isoflurane before euthanasia by cervical dislocation (lines 419-421) and in the corresponding animal models (lines 545 and 568).
2- So authors used cervical dislocation technique without anesthesia? this is not permittable. How the ethical committee approved this protocol?
All procedures were performed under the supervision of the animal facility personnel and, as previously stated, following European regulations. All protocols were revised and approved by the Ethics Committee in Experimental Research of the University of Valencia, and authorized by the Valencian Government, Spain) (2017/VSC/PEA/00151, September 21st, 2017; 2019/VSC/PEA/0137, June 19th, 2019; 2019/VSC/PEA/0138, June 19th, 2019). According to Sapnish regulations (BOE-A-2013-1337), cervical dislocation is a permitted method to euthanize rodents below 150g. Nevertheless, we routinely anesthetize mice with 4-5% isoflurane in a SomnoSuite (Kent Scientific, Torrington, CT, USA).
3- The method and software used for quantification in WB analysis are still not enough to allow reproducibility of the methods
RESPONSE: We apologize for not having explained with sufficient detail the band quantification and understand the necessity of this, given that two different methods using ImageJ have been described, depending on the version. In this new version of the manuscript we have specified the exact software used, which is the image processing package distribution of ImageJ2 known as Fiji, which has an automatic update function and bundles a lot of plugins that facilitate scientific image analysis and offers comprehensive documentation. A reference has been included: Schindelin, J., Arganda-Carreras, I., Frise, E., Kaynig, V., Longair, M., Pietzsch, T., … Cardona, A. (2012). Fiji: an open-source platform for biological-image analysis. Nature Methods, 9(7), 676–682. doi:10.1038/nmeth.2019.
The detailed quantification method is as follow:
The image was captured using an AutoChemi image analyser and the LabWorks 4.6 image acquisition software (UVP Inc., Upland, CA). The image was uploaded to the Fiji downloads (Windows 32-bit) platform. Then in the Image menu the type 32bit option was selected. Then, Adjust Brightness and contrast tool was selected, to obtain an image with less background. Next, the commands in this submenu to analyze one-dimensional electrophoretic gel was selected. The rectangular selection tool was used to outline all the bands of the protein of interest (determined by the Kd size, horizontal rectangle). From the Gel menu, First Lane (1) command is selected and the rectangle is labeled as ‘Lane 1’. Next the command Plot lanes is selected and the plot of the optical density opens up. With the Straight Line Selection Tool straight lines are drawn so that each peak of interest (corresponding to each band) defines a closed area. Then using the Wand tool from the bar menu, by clicking inside the peak the area under the curve is quantified as a measurement of the optical density. This process is repeated for each protein of interest and the internal loading controls, beta-actin or GAPDH. Data were transferred to an excel book and the ratio intensity protein of interest band/intensity of internal loading band calculated.
We have summarized the above explanation in the Methods section (lines 492-502). We truly hope that this will clarify the method used in a satisfactory manner. According to the reference aforementioned, the method used is accepted.
4 - Results should be consistent in number of groups. Neglection of one group due to any reason is not accepted
RESPONSE: We would like to express our apologies if our previous response was perceived by the reviewer as disrespectful, not at all our intention, and we regret any misunderstanding that it could have caused. We would like to point out that not being consistent with the number of groups was never intended to be due to neglect or lack of rigor. We had already determined in a previously published report (DOI: 10.1093/gerona/glw100) that osteostatin by itself did exert no effect on cytokine release nor NFkappaB activation by non-stimulated osteoblasts. In addition, prior literature (DOI:10.1302/2046-3758.71.BJR-2016-0242.R2) regarding the antioxidant effect of osteostatin did not presented its effect in non-stimulated osteoblastic cells. By our previous experience studying redox cell signaling and cell viability, we were aware that different molecules could exert effect on basal cells. Therefore, in our opinion, having one more group (Blank of osteostatin) in these experiments provided information of interest. We truly hope that the reviewer finds this explanation sufficiently satisfactory, since we acknowledge the relevance of the concern.
5- Authors mentioned they applied every possible comparison in methods. OK, but did you already applied the criteria in assessing the results? as no changes were made in figures.
RESPONSE: We had reanalyzed the data in the previous version, in response to the previous comments, to express the results as Mean +/- SD. However, we had only indicated the significance of the Control group versus naïve group, and osteostatin treated groups versus Control. We have now reanalyzed the data a second time, as suggested, and then added the significance between naïve and osteostatin treatments when p<0.05. We are grateful to the reviewer for this suggestion because we believe that by doing so the quality of data presentation has improved.
Reviewer 2 Report
Comments and Suggestions for Authors
We appreciate the authors' extensive revisions, which significantly improved the manuscript's quality. They have successfully addressed all concerns raised by the reviewers, and we agree that the manuscript is now acceptable for publication in its current form.
Author Response
We thank the reviewers for their constructive criticisms that have help us to improve the quality of the manuscript and the presentation of our work.
Round 3
Reviewer 1 Report
Comments and Suggestions for Authors
Thanks for the corrections
Comments on the Quality of English LanguageFine